# Deciphering the Chemical Fingerprint of *Astragalus membranaceus*: Volatile Components Attractive to *Bruchophagus huonchili* Wasps

**DOI:** 10.3390/insects14100809

**Published:** 2023-10-12

**Authors:** Chaoran Zhang, Penghua Bai, Jie Kang, Tian Dong, Haixia Zheng, Xianhong Zhang

**Affiliations:** 1College of Plant Protection, Shanxi Agricultural University, Jinzhong 030801, China; zhangcr9627@163.com (C.Z.); kangjie18834007528@163.com (J.K.); 18234475698@163.com (T.D.); zhenghaixia722@163.com (H.Z.); 2Institute of Plant Protection, Tianjin Academy of Agricultural Sciences, Tianjin 300384, China; baipenghua@126.com

**Keywords:** seed wasps, *Astragalus membranaceus* pods, plant volatiles, attractants

## Abstract

**Simple Summary:**

*Bruchophagus huonchili* is a serious threat to the yield and quality of the Chinese medicinal herb *Astragalus membranaceus* seeds. Plant volatile organic compounds (VOCs) are essential signaling substances for herbivorous insects when searching for host plants and mating partners. In this study, we collected and identified the VOCs emitted from *A. membranaceus* pods and evaluated the attraction of individual and combined compounds to *B. huonchili*. Our results showed that specific concentrations of five VOCs (cis-β-ocimene, hexyl acetate, hexanal, decanal, and β-caryophyllene) attract *B. huonchili* wasps. Moreover, our findings revealed that three formulations exhibited strong attraction to *B. huonchili* in Y-tube olfactometer assays and in the field. This study suggests that a blend with specific VOCs and ratios emitted from *A. membranaceus* could be used as attractants in traps for *B. huonchili* monitoring in the field.

**Abstract:**

*Bruchophagus huonchili* is a pest that poses a serious threat to the yield and quality of *Astragalus membranaceus* seeds. In this study, we employed solid-phase microextraction (SPME) and gas chromatography-mass spectrometry (GC-MS) to identify volatile organic compounds (VOCs) in *A. membranaceus* pods during the pod-filled period. Additionally, we utilized a Y-tube olfactometer to measure the behavioral response of *B. huonchili* to different individual VOCs and specific VOC-based formulations. The most effective formulations were further evaluated for their effectiveness in attracting wasps in the field. Our findings revealed that *A. membranaceus* pods emit 25 VOCs, including green leaf volatiles (GLVs) and terpenoid and aromatic compounds. Among these compounds, five were found to be most attractive to *B. huonchili* at the following concentrations: 10 µg/µL cis-β-ocimene, 500 µg/µL hexyl acetate, 100 µg/µL hexanal, 1 µg/µL decanal, and 10 µg/µL β-caryophyllene, with respective response rates of 67.65%, 67.74%, 65.12%, 67.57%, and 66.67%. In addition, we evaluated 26 mixed VOC formulations, and three of them were effective at attracting *B. huonchili*. Furthermore, field experiments showed that one of the formulations was significantly more effective than the others, which could be used for monitoring *B. huonchili* populations.

## 1. Introduction

The genus *Bruchophagus* (Hymenoptera: Eurytomidae) contains five species that negatively affect the seeds of the Chinese medicinal herb *Astragalus membranaceus*. One of them, *Bruchophagus huonchili* (Liao et Fan), has become a significant threat to the yield and quality of *A. membranaceus* seeds in recent years. Adult female wasps lay their eggs beneath the seed coat inside the pods, and the hatched larvae feed on the seeds, resulting in a consistent annual seed damage rate of 25% [1]. The difficulty in detecting adult wasp oviposition and larval damage makes chemical control a challenging task, and there are currently no registered pesticides available for controlling *B. huonchili* (http://www.chinapesticide.org.cn/ (accessed on 11 June 2023)). Moreover, the application of pesticides poses a risk of pesticide residue, which may affect the medicinal value of *A. membranaceus*. Therefore, finding an effective, feasible, and environmentally friendly measure to manage *B. huonchili* wasps is crucial for the sustainable development of the *A. membranaceus* industry.

Plant volatile organic compounds (VOCs) are essential signaling substances for herbivorous insects in searching for and locating their hosts, influencing insect host location, feeding, oviposition, mating, and other behaviors [2,3,4,5]. For example, female and male wasps of *Cotesia glomerata* (L.), *Cotesia marginiventris* (Cresson), *Microplitis rufiventris* (Kokujev), and *Microplitis mediator* (Haliday) were strongly attracted to herbivore-induced plant volatiles [6]. Also, *Bruchophagus roddi* (Gussakovsky) is a monophagous pest of alfalfa that attacks developing seeds and can recognize the host–plant volatile hexyl acetate antennally, which in turn stimulates behavioral activity [7]. The pea volatile cis-β-ocimene strongly attracts mated *Cydia nigricana* (Fabricius) females, which shows the importance of this VOC in their host location process [8].

Volatiles emitted by plants are usually diverse and complex blends [9,10,11]. The perception of blends of plant volatiles, rather than their individual components, plays a pivotal role in herbivorous insects during host-plant recognition [12,13]. Moreover, the volatiles released by plants, which encompass varied components, and their respective proportions (essentially the plant’s chemical fingerprint), which can lure herbivorous insects, are pivotal in the realm of pest surveillance and management [13,14,15,16,17,18]. For instance, a lure made from a mixture of 300 μg/μL of 2-hexanal and 180 μg/μL benzaldehyde had a high trapping success for *Callosobruchus chinensis* (Linnaeus) in the field [18]. Anfora et al. [16] found that a six-component synthetic lure, which approximated the ratio of components released by two grape varieties (Trebbiano and Sangiovese), was attractive and stimulated oviposition of the grapevine moth *Lobesia botrana* (Denis and Schiffermüller).

*Bruchophagus huonchili* is an oligophagous pest that specifically targets and feeds on *A. membranaceus* seeds, initiating oviposition within the seed pods during the pod-filled period. The exact mechanism that these wasps employ to locate *A. membranaceus* pods for oviposition remains, however, a mystery. Gaining insight into the plant’s chemical fingerprint and the volatiles that lure these wasps is essential for devising efficient pest control strategies. Our study aims to explore the behavioral response of *B. huonchili* wasps to various plant volatiles and component formulations, potentially contributing to the creation of effective and eco-friendly strategies for monitoring *B. huonchili*. By uncovering the specific volatile compounds and their optimal combinations that guide the wasps to their host plants, targeted and environmentally friendly control measures can be developed to reduce crop damage and improve agricultural productivity. Moreover, understanding the mechanisms and key factors involved in the attraction and oviposition behavior of *B. huonchili* wasps holds significant implications for sustainable pest management.

In this study, we employed solid-phase microextraction (SPME) and gas chromatography–mass spectrometry (GC-MS) to collect and identify the volatiles released by green *A. membranaceus* pods, aiming at revealing their chemical fingerprint. We used a Y-tube olfactometer to gauge the attractiveness of individual and combined VOCs to *B. huonchili*. Field trapping experiments were carried out to pinpoint the volatile components and their proportions that hold the greatest allure for *B. huonchili* wasps. Our study aims to identify the volatile components and their proportions that have a strong allure for these insects. The findings of this study are expected to contribute to the broader understanding of plant-insect interactions and provide valuable insights for the study of other pest species and the development of innovative pest control approaches.

## 2. Materials and Methods

### 2.1. Insects and Plants

*Astragalus membranaceus* has been growing for three years at Wujiabao Village of Taigu District, Dahuhui Village in Wuzhai County and Zhuangwanggou Village in Jing Le County, Shanxi Province. The pods of *A. membranaceus* were collected from the field during the pod-filling period for olfactometer experiments with *B. huonchili* wasps and volatile extraction (Figure 1).

During peak occurrence (June to August 2021), *B. huonchili* wasps were obtained from *A. membranaceus* pods in the experimental farmlands. The emerged adult wasps were reared indoors on honey water at a temperature of 25 ± 3 °C and 50–60% humidity. Newly emerged wasps (0–24 h) were isolated and identified as *B. huonchili* according to key characteristics [19]. Observations were made daily for adult emergence and the emerged adults were used in Y-tube olfactometer bioassays.

### 2.2. Volatile Compound Extraction and Identification

Solid-phase microextraction (SPME) was used to collect volatiles from fresh *A. membranaceus* pods at the pod filling period collected from the field [18]. A total weight of 2 g of *A. membranaceus* pods was placed in a small glass bottle (8 × 20 cm) into which purified air entered through a Teflon tube. Room air was pumped through activated charcoal into the glass bottle at a flow rate of 350 mL/min. Air was pulled out of the bottle through a trapping filter containing 300 mg of 80 mesh Porapak-Q adsorbent (Waters Corporation, Milford, MA, USA) and entered an activated charcoal filter on the opposite side of the bottle. Volatile compounds were collected in the Porapak-Q traps for 4 h and eluted with 1 mL of GC-grade hexane at room temperature under natural light conditions. There were three replicates. After each collection, the samples were stored at −80 °C before analysis [20].

Volatile extracts were analyzed by GC-MS using a Trace ISQ (Thermo Fisher Scientific, Waltham, MA, USA) equipped with a DB-5MS column (Agilent Technologies, Santa Clara, CA, USA) and coupled to 5975C MS. Helium was used as a carrier gas at 1.00 mL/min. The oven temperature was 50 °C for 5 min and then rose to 270 °C at a rate of 5 °C/min. The final temperature was held for 5 min. The components of the collected volatiles were identified by aligning their mass spectra with the benchmark compounds from the NIST2011 library and the quantification was evaluated using the peak area normalization method [20].

### 2.3. Compounds Tested

Eight out of the twenty-five volatile compounds found in *A. membranaceus* pods were selected to test for their attractiveness to *B. huonchili* wasps (Table 1). Test solutions were prepared at concentrations of 1, 10, 100, and 500 µg/µL, using paraffin oil (Sangon Biotech Co., Ltd., Shanghai, China) as the solvent.

### 2.4. Behavioral Assays

#### 2.4.1. Single-Component Behavioral Assay

To determine the attractiveness of individual volatiles from green *A. membranaceus* pods to *B. huonchili* wasps, a Y-tube olfactometer with a common glass tube (16 cm long with 20 mm diameter) and two lateral glass arms (10 cm long and 20 mm in diameter) (Bio-Olfactometer Company, Beijing, China) was used. Before the experiment, the olfactometer and silicone tubing were cleaned with 75% ethanol and dried with an air blower. The Y-tube olfactometer, gas flowmeter (Beijing Institute of Labor Protection, Beijing, China), filter bottle containing distilled water and activated carbon, and air sampler were connected in sequence using silicone tubing. The device was powered by an air pump, and the air passed through activated carbon and distilled water for filtration and humidification. The airflow in each arm was adjusted to 300 mL/min.

Before assessing the attraction of individual and combined VOCs to *B. huonchili*, a control experiment was conducted. In the control experiment, 10 μL of paraffin oil was applied to a filter paper strip and placed in each arm. The results showed that *B. huonchili* wasps distributed equivalent numbers in each arm in the absence of plant volatiles, which demonstrated that the Y-tube olfactometer was suitable for measuring the response rates of *B. huonchili* to different volatiles and component formulations.

The concentrations of the eight VOCs tested were 1, 10, 100, and 500 µg/µL diluted by paraffin oil. During the test, 10 μL of the VOCs were applied to a filter paper strip and placed in one of the testing arms, while a filter paper strip containing the same volume of paraffin oil was placed in the other arm as a control. *Bruchophagus huonchili* wasps (1 day old) were released individually at the base of the common arm of the Y-tube. The insect’s choice between the test arm and the control arm was observed and recorded. An insect was considered to have chosen the odor source or the control if it entered the respective arm and stayed there for more than 30 s within 5 min. If the insect did not make a choice within 5 min, it was considered unresponsive to the volatile. The filter paper strips were changed after every two insects were tested, and the position of the odor sources was then switched. Each concentration was repeated at least thirty times for wasps choosing either of the arms. The olfactometer was cleaned with ethanol, rinsed with distilled water, and placed in a 180 °C oven for 2 h to eliminate interference when changing different VOCs and concentrations.

The percent response rate of *B. huonchili* wasps was calculated as follows: 100% × [(wasps that chose the treatment arm)/(wasps that chose the treatment arm + wasps that chose the control arm)] [21].

#### 2.4.2. Multi-Component Behavioral Assay

To investigate the attraction of multi-component formulations to *B. huonchili* wasps, we selected five VOCs and concentrations with high attraction, which included 10 µg/µL cis-β-ocimene, 500 µg/µL hexyl acetate, 100 µg/µL hexanal, 1 µg/µL decanal, and 10 µg/µL β-caryophyllene, diluted by paraffin oil. The contents and proportion of each multi-component formulation were made according to the GC-MS results (Table 2). The total volume of each formulation was 10 μL and 10 μL of paraffin oil was used as a control. The behavioral assay method used was the same as described in Section 2.4.1.

### 2.5. Field Trapping Experiment

The three most attractive formulations of *B. huonchili* in the Y-tube assays (Formulations 6, 10, and 20; see Section 3) were selected and tested for insect attraction in two fields of *A. membranaceus* located in Huhui Village, Wuzhai County, and Wanggou Village, Jingle County. Yellow sticky traps baited with lure cores (Pherobio Technology Co., Ltd, Beijing, China) were used in the experiment. During the experiment, prepared lure cores containing 20 µL of different attractants were placed in the center of yellow sticky traps as treatments, and lure cores made with an equal volume of liquid paraffin oil (Beijing Ruihengjun’an Technology Co., Ltd. Beijing, China) were used as controls. Yellow sticky traps were hung in the field on 17 July and 23 July 2021, with the traps placed 10 cm above the top of the *A. membranaceus* plants and at a distance of 20 m between each other (Figure 2). Yellow sticky traps were collected on 22 and 28 July 2021. The wasps on the sticky traps were identified and the number of trapped wasps was counted. Each treatment was repeated three times.

### 2.6. Data Analyses

We used SPSS version 17.0 (SPSS, Chicago, IL, USA) statistical software for all data analyses. The data on the percent response rate of *B. huonchili* wasps in the Y-tube olfactometer bioassay at different concentrations of each stimulus were analyzed using Chi-square (*χ*^2^) tests. The number of *B. huonchili* wasps captured on yellow sticky traps is presented as the mean ± standard error and was analyzed by one-way ANOVA, followed by the Tukey multiple comparison test to compare between the control and the three different VOC formulations, with the level of significance set at *p* < 0.05.

## 3. Results

### 3.1. Volatile Compound Extraction and Identification

In this study, 25 volatile compounds were identified from *A. membranaceus* pods during the pod-filling period [22], including green leaf volatiles (GLVs) and terpenoid and aromatic compounds (Table 3). For example, GLVs, such as hexyl acetate and hexanal, were detected in the volatile blends. Cis-β-ocimene and terpinolene, two common terpenoid volatiles, were also identified from the pods. From these results, eight volatiles were selected for Y-tube behavioral assays, which included ethyl palmitate, isopentyl isovalerate, cis-β-ocimene, 1-octen-3-ol, hexyl acetate, hexanal, decanal, and β-caryophyllene. These compounds were selected because they were the major components of the blend, constituting about 67% of the emitted volatiles.

### 3.2. Behavioral Assays

#### 3.2.1. Single-Component Behavioral Assay

The results showed that 10 µg/µL cis-β-ocimene, 500 µg/µL hexyl acetate, 100 µg/µL hexanal, 1 µg/µL decanal, and 10 µg/µL β-caryophyllene attracted 67.65% (*χ*^2^ = 4.235, *p* = 0.040), 67.74% (*χ*^2^ = 3.903, *p* = 0.048), 65.12% (*χ*^2^ = 3.903, *p* = 0.047), 67.57% (*χ*^2^ = 4.568, *p* = 0.033), and 66.67% (*χ*^2^ = 4.000, *p* = 0.046) of *B. huonchili* wasps, respectively (Figure 3A–E). However, 1 µg/µL (*χ*^2^ = 10.800, *p* = 0.001) and 10 µg/µL (*χ*^2^ = 7.529, *p* = 0.006) hexyl acetate, 500 µg/µL 1-octen-3-ol (*χ*^2^ = 14.286, *p* < 0.001), and isoamyl isovalerate (*χ*^2^ = 9.000, *p* = 0.003) exhibited repellent effects on *B. huonchili* wasps (Figure 3B,F,H). In addition, the four concentrations of ethyl palmitate had no effects on *B. huonchili* wasps (*p* > 0.05) (Figure 3G).

#### 3.2.2. Multi-Component Behavioral Assay

Out of the 26 formulations tested (see Table 2), *B. huonchili* wasps were attracted to formulations 6, 10, and 20, with the highest percent attraction of 73.07% for formulation 6 (*χ*^2^ = 7.410, *p* = 0.006), followed by 66.66% attraction for formulation 10 (*χ*^2^ = 4.333, *p* = 0.037), and 65.12% attraction for formulation 20 (*χ*^2^ = 3.930, *p* = 0.047) (Figure 4). None of the other formulations had an effect on *B. huonchili* behavior (*p* > 0.05).

### 3.3. Field Trapping Experiment

In the field, formulation 6 showed the highest wasp-attracting effect, capturing fourteen times and seventeen times more wasps/trap than the controls on 22 and 28 July, respectively, in the field of Huhui Village and nine times and ten times more wasps/trap than the controls on 22 and 28 July, respectively, in field of Wanggou Village (Table 4). The second best formulation was formulation 20, with wasp-attracting effects 6–11 times greater than the controls (Table 4). There was no significant difference in attractiveness between formulations 20 and 10 (Table 4).

## 4. Discussion

Plant volatiles have a unique chemical fingerprint composed of specific concentrations and ratios, which plays a crucial role in their life activities, such as long-distance host searching, nutrition supplementation, mating, and oviposition, of herbivorous insects [3,4,23,24]. Solid-phase microextraction (SPME) and gas chromatography–mass spectrometry (GC-MS) are widely used methods for extracting and identifying these plant volatiles. For instance, Jing et al. [25] successfully analyzed the herbivore-induced plant volatiles emitted from tea leaves and induced by *Ectropis obliqua* (Prout) using these methods, and Zhang et al. [26] found that the middle and lower leaves of *Nicotiana benthamiana* (Linnaeus) emitted higher amounts of (*Z*)-3-hexen-1-ol than the upper leaves using SPME and GC-MS. In this study, a total of 25 VOCs were identified from *A. membranaceus* pods during the drum pod-filled period, including GLVs, such as hexanal, which are also detected in the pods of *Vigna radiate* (Wilczek) and soybean (*Glycine max* (Linnaeus)) [18,27]. In addition, cis-β-ocimene and terpinolene, two common terpenoid volatiles, were identified from the *A. membranaceus* pod headspace samples. GLVs and terpenoid volatiles are commonly emitted from healthy plants and herbivore-induced plants, which can provide important signals in plant-herbivore interactions [3,4].

Plant volatiles can have either attractive or repellent effects on insects, with the effectiveness depending on their concentration. For instance, Li et al. [28] reported that *Microplitis mediator* (Haliday) is strongly attracted to 10 µg/µL β-ionone, but when the concentration reached 100 µg/µL, β-ionone showed repellency to male wasps. Similarly, compared with the control, *Chouioia cunea* Yang was not attracted to nonadecane at high and low doses, whereas it was more attracted to nonadecane at an intermediate dose [21]. Also, *Aphidius colemani* (Viereck) had a significant preference for benzaldehyde at 50 ng and 10 ng doses and a negative response at a higher dose of 1 μg [29]. In our study, we identified the optimal concentrations of eight VOCs from *A. membranaceus* pods that are attractive to *B. huonchili* wasps. These VOCs were commonly found in these host plants and can be purchased from commercial suppliers, making them suitable as potential attractants. Among the eight VOCs, five were found to be most attractive to *B. huonchili* at the following concentrations: 10 µg/µL cis-β-ocimene, 500 µg/µL hexyl acetate, 100 µg/µL hexanal, 1 µg/µL decanal, and 10 µg/µL β-caryophyllene, with respective response rates of 67.65%, 67.74%, 65.12%, 67.57%, and 66.67%.

Although a single VOC could attract *B. huonchili* wasps, plants usually emit complex VOC blends and perception of these blends, rather than their individual components, by herbivorous insects plays a pivotal role in host plant recognition [9,10,11,12,13]. Based on the GC results, the optimal ratio of hexyl acetate:hexanal:decanal:β-caryophyllene was 521:128:23:17 (formulation 6), which showed strong effects on *B. huonchili* behavior, reaching 73.03% attraction. The attraction of *B. huonchili* wasps to formulation 6 was higher than to each of the single components. Similarly, Zhu et al. [30] showed that a 1:22:5 mixture of ethanol, acetic acid, and phenethyl alcohol from ripe mango is more attractive to *Drosophila melanogaster* (Meigen) than any of the single components. A 10:24:6:0.2 mixture of palm oil ester, linoleic acid ethyl ester, linoleic acid methyl ester, and linoleic acid from wheat bran fermentation products increased the attraction and oviposition of house flies, *Musca domestica* (L.), compared to each individual component [31].

In addition, many terpenoids emitted from plants have behavioral effects on insects. For instance, *Chelonus insularis* (Cresson) female parasitoids were attracted to the binary blend of α-pinene and α-copaene [32]. *Aphytis melinus* (DeBach) females showed positive chemotaxis toward d-limonene and cis-β-ocimene [33]. In our study, both Y-tube olfactometer assays and field experiments showed that the formulations 6 (hexyl acetate + hexanal + decanal + β-caryophyllene), 10 (β-caryophyllene + hexanal + decanal), and 20 (cis-β-ocimene + β-caryophyllene) were attractive to *B. huonchili* wasps. All three formulations contained terpenoid compounds, i.e., either cis-β-ocimene or β-caryophyllene, indicating their importance as host-plant attractants for *B. huonchili*.

In conclusion, we identified specific concentrations and ratios of the VOCs cis-β-ocimene, hexyl acetate, hexanal, decanal, and β-caryophyllene from *A. membranaceus* pods that are attractive to *B. huonchili* wasps. Our study revealed three multi-component formulations (formulations 6, 10, and 20) containing five *A. membranaceus* VOCs that attracted *B. huonchili* in Y-tube olfactometer assays and in the field. One of them (formulation 6, which is composed of hexyl acetate, hexanal, decanal, and β-caryophyllene) showed to be the most attractive of the three formulations, making it a good candidate for use as a lure for monitoring *B. huonchili* wasps.

## Figures and Tables

**Figure 1 insects-14-00809-f001:**
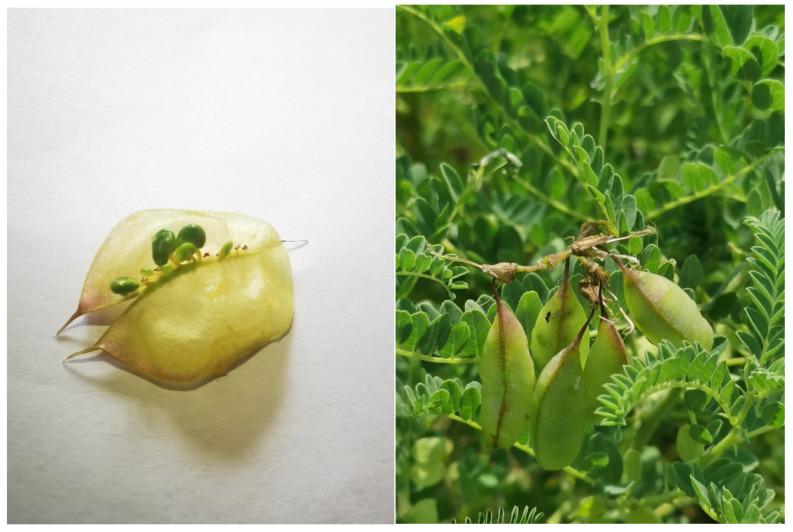
The pods of *Astragalus membranaceus* during pod filling period in the field.

**Figure 2 insects-14-00809-f002:**
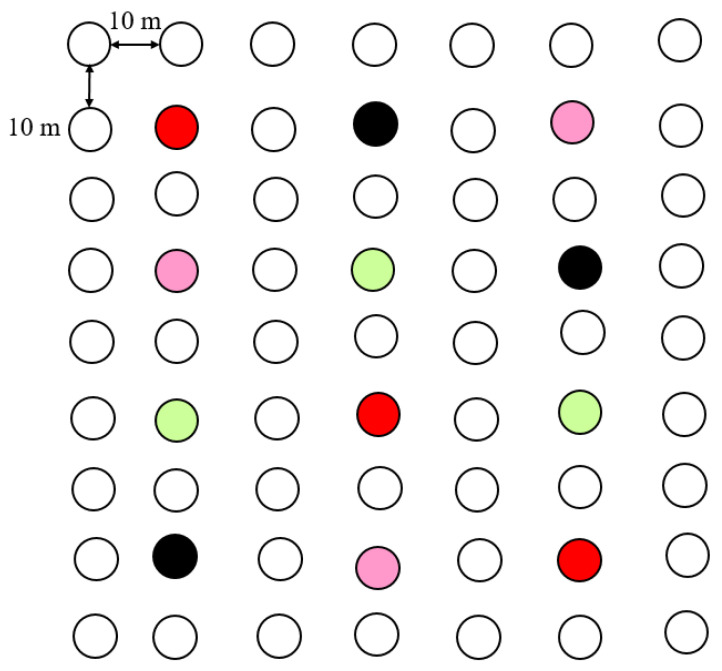
Design of field experiment to test the attraction of *Bruchophagus huonchili* to different formulations of volatiles. The red circles represent the traps baited with formulation 6, the purple circles represent the traps baited with formulation 10, the green circles represent the traps baited with formulation 20, and the black circles represent the control traps. The distance between the two lure cores was 20 m, and each treatment was replicated three times.

**Figure 3 insects-14-00809-f003:**
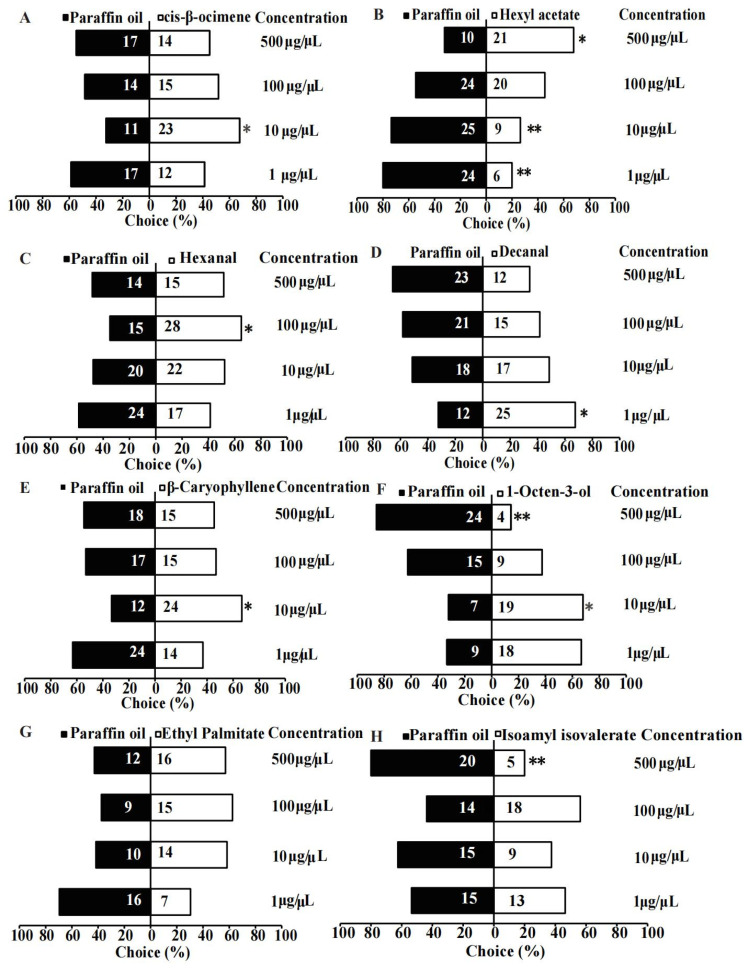
Responses of *Bruchophagus huonchili* wasps to eight volatiles emitted from *Astragalus membranaceus* pods in a Y-tube olfactometer. (**A**) Cis-β-ocimene, (**B**) Hexyl acetate, (**C**) Hexanal, (**D**) Decanal, (**E**) β-Caryophyllene, (**F**) 1-Octen-3-ol, (**G**) Ethyl Palmitate, and (**H**) Isoamyl isovalerate attraction. Volatiles were tested at concentrations of 1 μg/μL, 10 μg/μL, 100 μg/μL, and 500 μg/μL. * indicates a significant difference compared with the control group as analyzed by *χ*^2^ tests (*p* < 0.05), ** indicates *p* < 0.01.

**Figure 4 insects-14-00809-f004:**
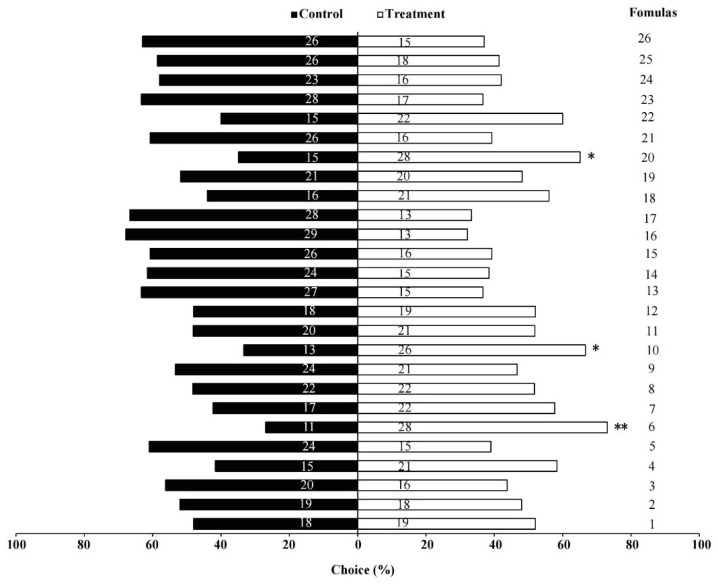
Responses of *Bruchophagus huonchili* wasps to multi-component volatile formulations. * indicates a significant difference compared with the control group as analyzed by *χ*^2^ tests (*p* < 0.05), ** indicates *p* < 0.01.

**Table 1 insects-14-00809-t001:** List of standard compounds tested for *Bruchophagus huonchili* attraction.

Standard Compound	CAS	Purity (%)	Source
Ethyl palmitate	628-97-7	98	Shanghai Macklin Biochemical Co., Ltd. (Shanghai, China)
Isoamyl isovalerate	659-70-1	98	Shanghai Macklin Biochemical Co., Ltd. (Shanghai, China)
Cis-β-ocimene	13877-91-3	98	Shanghai Macklin Biochemical Co., Ltd. (Shanghai, China)
1-Octen-3-ol	3391-86-4	98	Shanghai Aladdin Biochemical Technology Co., Ltd. (Shanghai, China)
Hexyl acetate	142-92-7	99	Shanghai Macklin Biochemical Co., Ltd. (Shanghai, China)
Hexanal	66-25-1	97	Shanghai Macklin Biochemical Co., Ltd. (Shanghai, China)
Decanal	112-31-2	97	Shanghai Macklin Biochemical Co., Ltd. (Shanghai, China)
β-Caryophyllene	87-44-5	80	Shanghai Macklin Biochemical Co., Ltd. (Shanghai, China)

**Table 2 insects-14-00809-t002:** Composition of the 26 formulations tested for *Bruchophagus huonchili* attraction.

Formula No.	Composition	Content (µL)
1	Cis-β-ocimene + Hexyl acetate + Hexanal + Decanal + β-Caryophyllene	26 + 521 + 128 + 23 + 17
2	Cis-β-ocimene + Hexyl acetate + Hexanal + Decanal	26 + 521 + 128 + 23
3	Cis-β-ocimene + Hexyl acetate + Hexanal + β-Caryophyllene	26 + 521 + 128 + 17
4	Cis-β-ocimene + Hexyl acetate + Decanal + β-Caryophyllene	26 + 521 + 23 + 17
5	Cis-β-ocimene + Hexanal + Decanal + β-Caryophyllene	26 + 128 + 23 + 17
6	Hexyl acetate + Hexanal + Decanal + β-Caryophyllene	521 + 128 + 23 + 17
7	Cis-β-ocimene + Hexyl acetate + Hexanal	26 + 521 + 128
8	Cis-β-ocimene + Hexyl acetate + Decanal	26 + 521 + 23
9	Cis-β-ocimene + Hexyl acetate + β-Caryophyllene	26 + 521 + 17
10	Cis-β-ocimene + Hexanal + Decanal	26 + 128 + 23
11	Cis-β-ocimene + Hexanal + β-Caryophyllene	26 + 128 + 17
12	Cis-β-ocimene + Decanal + β-Caryophyllene	26 + 23 + 17
13	Hexyl acetate + Hexanal + Decanal	521 + 128 + 23
14	Hexyl acetate + Hexanal + β-Caryophyllene	521 + 128 + 17
15	Hexyl acetate + Decanal + β-Caryophyllene	521 + 23 + 17
16	Hexanal + Decanal + β-Caryophyllene	128 + 23 + 17
17	Cis-β-ocimene + Hexyl acetate	26 + 521
18	Cis-β-ocimene + Hexanal	26 + 128
19	Cis-β-ocimene + Decanal	26 + 23
20	Cis-β-ocimene + β-Caryophyllene	26 + 17
21	Hexyl acetate + Hexanal	521 + 128
22	Hexyl acetate + Decanal	521 + 23
23	Hexyl acetate + β-Caryophyllene	521 + 17
24	Hexanal + Decanal	128 + 23
25	Hexanal + β-Caryophyllene	128 + 17
26	Decanal + β-Caryophyllene	23 + 17

**Table 3 insects-14-00809-t003:** Volatile compounds and their relative amounts emitted from *Astragalus membranaceus* pods.

Chemicals	CAS	Retention Tim (min)	Relative Content (%)
Cis-β-ocimene	13877-91-3	14.03	12.49
Hexyl acetate	142-92-7	13.3	11.96
1-Octen-3-ol	3391-86-4	12.11	10.11
Hexanal	66-25-1	5.92	9.37
Decanal	112-31-2	20.54	9.21
β-Caryophyllene	87-44-5	27.78	8.75
Ethyl palmitate	628-97-7	43.95	8.06
Isoamyl isovalerate	659-70-1	14.11	6.17
2-Methylbutyl2-methylbutyrate	2445-78-5	16.73	3.45
1,4-Dichlorobenzene	106-46-7	13.36	3.29
Hexamethylcyclotrisiloxane	541-05-9	6.34	2.22
2-Pentylfuran	3777-69-3	12.44	1.97
Naphthalene	91-20-3	19.74	1.73
8-Fluoro-1-octanol	111-87-5	15.57	160
Ethanol	64-17-5	5.92	1.48
(*S*)-(-)-2-Methyl-1-butanol	1565-80-6	4.46	1.07
Methyl heptanoate	106-73-0	14.31	1.07
Tetradecamethylcycloheptasiloxane	107-50-6	29.24	0.99
1-Heptanol	111-70-6	15.03	0.95
Cyclohexene	110-83-8	16.05	0.82
Benzyl alcohol	100-51-6	99	0.78
Leaf alcohol	928-96-1	7.63	0.70
Methyl isovalerate	556-24-1	2.81	0.66
Methyl hexanoate	106-70-7	10.96	0.58
Octamethylcyclotetrasiloxane	556-67-2	12.18	0.53

**Table 4 insects-14-00809-t004:** Effects of three host-plant volatile formulations on *Bruchophagus huonchili* captures in the field.

Formulation No.	Mean ± SE Number of Wasps Captured in Huhui Village ^1^		Mean ± SE Number of Wasps Captured in Wanggou Village ^1^	
	22 July	28 July	22 July	28 July
6	171.67 ± 7.80 ^a^	202.67 ± 8.74 ^a^	77.33 ± 7.51 ^a^	83.33 ± 6.49 ^a^
10	115.33 ± 9.96 ^b^	124.33 ± 16.18 ^b^	44.67 ± 1.20 ^b^	51.33 ± 4.98 ^b^
20	138.67 ± 5.21 ^b^	105.67 ± 9.94 ^b^	48.67 ± 2.60 ^b^	50.00 ± 2.52 ^b^
Control	12.33 ± 0.88 ^c^	11.67 ± 0.33 ^c^	8.67 ± 0.88 ^d^	8.33 ± 0.33 ^c^

^1^ Different letters within a column indicate significant differences as determined by one-way ANOVA followed by Tukey’s HSD tests (*p* < 0.05). Each treatment was repeated three times.

## Data Availability

The data presented in this study are available in this paper.

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
