# Peer review of "Deciphering the Chemical Fingerprint of Astragalus membranaceus: Volatile Components Attractive to Bruchophagus huonchili Wasps"

_insects, 2023, doi:10.3390/insects14100809_

Round 1

Reviewer 1 Report

The introduction and discussion reference a lot of instances of herbivore attraction to plants but there is no mention of the host location behavior of B. roddi and B. gibbus, which have been investigated. Are there other more pertinent examples of attraction of seed chalcids to host plants? How do the volatiles that you collected from Astragalus in your study compare to volatiles collected from other species? What was the species of Astragalus that you were working on? The introduction and discussion should be more focused on the seed chalcid-plant interactions you are studying, or at least confined to pests of fruiting structures of the Fabaceae. Additional light language editing is needed.

Here are some  additional comments and questions.

What species of Astragalus are you working on?

l. 9 - Bruchophagus sp. a mixed population of pests – do you mean an undescribed species? I assume you are talking about one species. If you are talking about several species, use spp. and omit mixed population as population refers to individuals of the same species.

l.14 – remove “the” before specific concentrations.

l. 17 – change sp. to spp. Throughout the manuscript wherever you mean more than one species.

Abstract

l. 21 – Change to “Bruchophagus spp. are pests that pose a serious….”

l. 25 – change to “which is the dominant species in the mixed populations of Bruchophagus spp…”

l. 31 – response rates are percentages. I assume this is relative to a control?

Introduction

l. 41 – I would change to “several species of Bruchophagus occur in mixed populations attacking Chinese medical herb Astragalus seeds…

l. 43 – The family for Habrocytus is Pteromalidae. Habrocytus is the genus name.

l. 51 – you switch to talking only about B. huonchili in the last line of the paragraph. Are all members of that genus a problem? If so, why concentrate specifically in that first paragraph on control of B. huonchili? Here and in l. 77, you might mention that B. huonchili is the most common pest. Perhaps you are doing the lab bioassays on this insect because it is the one you have in culture but you should be clear that you are concentrating on this species because it is the most important. I note that in l. 82, you finally mention that it is the dominant species in the mixture of species. You might make this statement at the beginning of the introduction. Line 43 would be a good place for this.

L. 93 – Can you tell which species the pupae are? How did you determine that you were collecting and experimenting with B. huonchili rather than another Bruchophagus species?

l. 97 – Were the three-year old plants from which you collected pods grown in the field? The next line (100) indicates that you collected the pods from the field. Please clarify.

l. 100 – Describe the volatile collection system or reference the procedure. What size chamber, how long the plants were left in the chamber, how many or what mass of plants were placed in the chamber, etc..

l. 107 – What methods did you use to tentatively identify the volatiles? Did you add an internal standard during volatile collection to try to quantify the volatiles?

l. 146 – “Prepare a 2 mL solution with paraffin oil as solvent”. Does this describe the preparation of the volatile stock solution? Please clarify.

l. 153 – “Yellow sticky traps with lure cores (The West Company, Phoenixville, 153 PA, USA)”. Please describe the traps better as I can’t find this company. Flat panels? What dimensions? How arrayed in the field? What is a lure core? How many traps per treatment were deployed in each of the three (or four) repetitions in each field?

l. 167 – Duncan’s multiple comparison test is rarely used any more. Is there a reason that was chosen? Tukey’s HSD might be better.

l. 172 – Add “tentatively” identify. What database was used for the identifications?

l. 173 - What is granulation stage? Earlier you called it the seed-swelling stage. L. 244 – you use the term “drum granule stage”. Please be consistent and perhaps cite a paper that describes the vegetative/reproductive stages of this plant.

Fig. 1 – what was the identity of the 25 volatiles found in the seed-swelling flower headspace? Please put in a table.

Fig. 2- how would you explain that high dosages of hexyl acetate (500 ug/uL) were attractive but low doses (1 and 10 ug/ul) were repellent)? That doesn’t make intuitive sense.

l. 200 – How did you decide on the specific concentrations of components within the multi-component mixtures?

l. 213 – at what growth stage were the Astragalus fields used in the field trials? Were they at the seed-swelling stage? If vegetative rather than reproductive, they may have been significantly less attractive than the baited traps. How might the traps compete in fields that have high numbers of attractive seed-swelling stage plants?

Table 3 – please put in the number of replicates and the meaning of the means separations letters.

Discussion – “This suggests that the 282 plant volatiles that attract these five species of Bruchophagus wasps are the same or similar, indicating relatively conserved olfactory genes of these seed wasps and potentially similar functions.” Without knowing the number of each species that you caught, and their true numbers in the field, it is difficult to know if you attracted the five species in similar proportions to their abundance.

l. 295 – “…B. huonchili wasps, in combination with sex pheromones”. Has the sex pheromone of this seed wasp been identified? Is there known attraction to a pheromone? If not, this is not really pertinent to the discussion. You discuss attraction of parasitoids by sex pheromone to herbivores but the species you are working on is an herbivore, not a parasitoid. I think the discussion should be more focused on your plant-insect or related plants and related pest insect systems.

Light editing is needed.

Author Response

Comments to the Author

1.The introduction and discussion reference a lot of instances of herbivore attraction to plants but there is no mention of the host location behavior of B. roddi and B. gibbus, which have been investigated. Are there other more pertinent examples of attraction of seed chalcids to host plants? How do the volatiles that you collected from Astragalus in your study compare to volatiles collected from other species? What was the species of Astragalus that you were working on? The introduction and discussion should be more focused on the seed chalcid-plant interactions you are studying, or at least confined to pests of fruiting structures of the Fabaceae. Additional light language editing is needed

Response: Added the seed chalcid-plant interactions as request in the introduction, Line in 50

Plant volatile organic compounds (VOCs) are essential signaling substances for herbivorous insects in searching for and locating their hosts, influencing insect host location, feeding, oviposition, mating, and other behaviors [2-5]. For example, female and male wasps of all four species, Cotesia glomerata (L.), Cotesia marginiventris (Cresson), Microplitis rufiventris Kokujev and Microplitis mediator, were strongly attracted by herbivore-induced plant volatiles [6]; Similarly, B. roddi is a monophagous pest of alfalfa that parasitizes developing seeds, which can recognized host plant volatile hexyl acetate by antennae to stimulate behavioral activity [7]. Pea plant volatile b-ocimene strongly attract the behaviour of mated Cydia nigricana females, which supports the importance of b-ocimene in host location [8].

Volatiles emitted by plants are usually diverse and complex blends [9-11]. Perception of blends of plant volatiles, rather than individual components, plays a pivotal role in host recognition [12-13] Moreover, the volatiles released by plants, which lure herbivorous insects (encompassing varied components and their respective proportions, essentially the plant's chemical fingerprint), are pivotal in the realm of pest surveillance and management [13-18]. For instance, a lure made from a mixture of 300 μg/μL 2-hexanal and 180 μg/μL benzaldehyde has a high trapping rate for Callosobruchus chinensis in the field [18]; Anfora et al. found that a six-component synthetic lure, which approximated the ratio of components released by two grape varieties (Trebbiano and Sangiovese), was attractive and stimulated oviposition of the grapevine moth Lobesia botrana [16].

References:

  1. Xu, H.; Gaylord, D.; Degen, T.; Zhou, G. X.; Laplanche, D.; Henryk, L.; Turling, T. C. J. Combined use of herbivore-induced plant volatiles and sex pheromones for mate location in braconid parasitoids.Plant , 2017. 40: 330-339.
  2. Light, D. M.; Kamm, J. A.; Buttery, R. G. Electroantennogram response of alfalfa seed chalcid,Bruchophagus roddi (Hymenoptera: Eurytomidae) to host- and nonhost-plant volatiles. Journal of Chemical Ecology. 1992. 18: 333-352.
  3. Gunda, T.;Rangar, H. R.; Helmut, S.; Geir, K. K. Pea plant volatiles guide host location behaviour in the pea moth. Arthropod-Plant Interactions. 2014.8:109-122.
  4. McCormick, A. C.; Unsicker, S. B.; Gershenzon, J. The specificity of herbivore-induced plant volatiles in attracting herbivore enemies. Trends in Plant Science. 2012. 17:303–310
  5. Visser, J. H. Host odor perception in phytophagous insects. Annual Review of Entomology. 1986. 31: 121–144.
  6. Schiestl, F. P. The evolution of floral scent and insect chemical communication. Ecology Letters. 2010. 13: 643–656.
  7. Bruce, T. J.; Pickett, J. A. Perception of plant volatile blends by her bivorous insects–finding the right mix. Phytochemistry. 2011. 72:1605–1611.

13、Xiu, C. L.; Pan, H. S.; Liu, B.; Luo, Z. X.; Williams, L.; Yang, Y. Z.; Lu, Y. H. Perception of and Behavioral Responses to Host Plant Volatiles for Three Adelphocoris Species. Journal of Chemical Ecology. 2019. 45: 779-788.

14     Kessler, A.; Baldwin, I. T. Defensive function of herbivore-induced plant volatile emission in nature. Science. 2001. 291, 2141-2144.

  1. Cha, D. H.; Nojima, S.; Hesler S. P.; Zhang, A.; Linn, C. E Jr,; Roelofs, W. L, Loeb, G. M. identification and field evaluation of grape shoot volatiles attractive to female grape berry moth (Paralobesia viteana). Journal of Chemical Ecology. 2008, 34, 1180-1189.
  2. Anfora, G.; Tasin, M.; De, Cristofaro. A.; Ioriatti, C.; Lucchi A. Synthetic grape volatiles attract mated Lobesia botrana females in laboratory and field bioassays. Journal of Chemical Ecology. 2009,35, 1054-1062.
  3. Oseiowusu, J.; Vuts, J.; Caulfield, J. C.; Woodcock, C. M.; Withall, D. M.; Hooper, A. M.; Birkett, M. A. Identification of semiochemicals from cowpea, Vigna unguiculata, for low-input management of the legume pod borer, Maruca vitrata. Journal of Chemical Ecology. 2020. 1: 1-11.
  4. 1 Wang, H. M.; Bai, P. H.; Zhang, J.; Zhang, X. M.; Hui, Q.; Zheng, H. X.; Zhang, X. H. Attraction of bruchid beetles Callosobruchus chinensis (L.) (Coleoptera: Bruchidae) to host plant volatiles. Journal of Integrative Agriculture. 2020. 19: 2-11.

Line in 270

In this study, a total of 25 volatile compounds were identified from Astragalus membranaceus pods during the drum pod filled period, including green leaf volatiles, such as hexanal, were detected in the pods of Vigna radiate and soybean (Glycine max) (18, 27). In addition, ocimene and terpinolene as common terpenoid volatiles were identified in pod volatiles. GLVs and terpenoid volatiles are common in healthy plants and herbivore-induced plants, which can provide important signals in plant-herbivore interactions [3,4].

References:

  1. O'Neill, B. F.; Zangerl, A. R.; Delucia, E. H.; Berenbaum, M. R. Olfactory preferences of Popillia japonica, Vanessa cardui, and Aphis glycines for Glycine max grown under elevated CO2. Environmental Entomology. 2010. 4: 1291-1301.

I working on Astragalus membranaceus.

  1. What species of Astragalus are you working on?

Response: The species of Astragalus was Astragalus membranaceus. The species of Astragalus was revised in the manuscript.

  1. Bruchophagus sp. a mixed population of pests – do you mean an undescribed species? I assume you are talking about one species. If you are talking about several species, use spp. and omit mixed population as population refers to individuals of the same species.

Response:

We focused on the species of Bruchophagus huonchili in our manuscript and have modified the mixed population of Bruchophagus spp. to B. huonchili according to the recommendations of the reviewers.

  1. remove “the” before specific concentrations.

Response:  Corrected as request, we have removed “the” before specific concentrations. Line in 14 in the revised manuscript.

  1. change sp. to spp. Throughout the manuscript wherever you mean more than one species.

Response: We focused on the species of Bruchophagus huonchili in our manuscript and have modified the mixed population of Bruchophagus spp. to B. huonchili according to the recommendations of the reviewers. Moreover, we changed ' sp. ' to ' spp.  ', Line 37 in the revised manuscript.

  1. Change to “Bruchophagus spp. are pests that pose a serious….”

Response: “Bruchophagus spp.” has been placed by “Bruchophagus huonchili.” in the revised manuscript.

  1. change to “which is the dominant species in the mixed populations of Bruchophagus spp…”

Response: Several species of Bruchophagus occur in mixed populations attacking Chinese medical herb A. membranaceus seeds, however, Bruchophagus huonchili poses a serious threat to the yield and quality of Astragalus seeds. Line 39 in the revised manuscript.

  1. response rates are percentages. I assume this is relative to a control?

Response: The response rate was calculated as follows: Response rate = 100% × (wasps that chose the treatment arm) / (wasps that chose the treatment arm + wasps that chose the control arm) [21]. The sentence was added in Materials and Methods on line 160 in the revised manuscript.

References:

  1. Li, M.; Yang, Y. X.; Yao,Y. H.; Xiang, W. F.; Han, J. Y.; Wang, Y. H.; Bai, P. H.; Wang, J.; Zhu, G. P.; Man, L.; Zhang, F.; Pan, L. Isolation and Identification of Attractants from the Pupae of Three Lepidopteran Species for the Parasitoid Chouioia Cunea Yang. Pest Management Science. 2020. 76 : 1920–1928.

I would change to “several species of Bruchophagus occur in mixed populations attacking Chinese medical herb Astragalus seeds…

Response: We focused on the species of Bruchophagus huonchili in our manuscript and have modified the mixed population of Bruchophagus spp. to B. huonchili according to the recommendations of the reviewers. Moreover, we changed ' sp. ' to ' spp.  ', Line 37 in the revised manuscript.

  1. The family for Habrocytus is Pteromalidae. Habrocytus is the genus name.

Response: We agree with your point. We changed ' Habrocytus ' to ' Pteromalidae. ' Line 39 in the revised manuscript.

  1. – you switch to talking only about B. huonchili in the last line of the paragraph. Are all members of that genus a problem? If so, why concentrate specifically in that first paragraph on control of B. huonchili? Here and in l. 77, you might mention that B. huonchili is the most common pest. Perhaps you are doing the lab bioassays on this insect because it is the one you have in culture but you should be clear that you are concentrating on this species because it is the most important. I note that in l. 82, you finally mention that it is the dominant species in the mixture of species. You might make this statement at the beginning of the introduction. Line 43 would be a good place for this.

Response: We agree with your point that our manuscript should focused on the species of Bruchophagus huonchili, because B. huonchili. is pest that poses a serious threat to the yield and quality of Astragalus seeds. Moreover, we have make this statement at the beginning of the introduction. Line 39 in the revised introduction.

  1. Can you tell which species the pupae are? How did you determine that you were collecting and experimenting with B. huonchili rather than another Bruchophagus species?

Response: The pupae of the species for behavioral assays were B. huonchili. Newly emerged wasps (0-24 h) were isolated and identify the species of B. huonchili according to the key characteristics [19]. Line 101 in the revised introduction.

13.Were the three-year old plants from which you collected pods grown in the field? The next line (100) indicates that you collected the pods from the field. Please clarify.

Response: Astragalus mongolicus Bge. has been growing for three years at Wujiabao Village of Taigu District, Dahuhui Village in Wuzhai County and Zhuangwanggou Village in Jing Le County, Shanxi Province. The pods of A. membranaceus were collected from the field during the pod filling period for B. huonchili wasps and volatile extraction (Figure 1).

  1. Describe the volatile collection system or reference the procedure. What size chamber, how long the plants were left in the chamber, how many or what mass of plants were placed in the chamber, etc..

Response: Solid phase microextraction was used to collect volatiles from fresh Astragalus mongolicus pods at the pod filling period collected from the field [18].A total weight of 2 g of A. membranaceus pods were placed in a small glass bottle (8 × 20 cm) into which purified air entered through a Teflon tube.Room air was pumped through activated charcoal into the conical flask at a flow rate of 350 mL/min. Air was pulled out of the bottle through a trapping filter containing 300 mg of 80 mesh Porapak-Q adsorbent (Waters Corporation, Milford, Massachusetts, USA), and entered an activated charcoal filter on the opposite side of the bag. Volatile compounds were collected in the Porapak Q traps for 4 h and eluted with 1 mL of GC-grade hexane at room temperature under natural light conditions. The three repetitions are based on biological repetition. After each collection, 1 mL of hexane and the eluate was stored at −80 ℃ before analysis [20]. Line 108 in the revised introduction.

  1. What methods did you use to tentatively identify the volatiles? Did you add an internal standard during volatile collection to try to quantify the volatiles?

Response: The components of the collected volatiles were identified by aligning their mass spectra with the benchmark compounds from the NIST2011 library and the quantification was evaluated using the peak area normalization method [18]. Line 119 in the revised introduction.

  1. “Prepare a 2 mL solution with paraffin oil as solvent”. Does this describe the preparation of the volatile stock solution? Please clarify.

Response: Change “Prepare a 2 mL solution with paraffin oil as solvent” to “The volume of each treatments was 10 μL and compared with 10 μL of paraffin oil as solvent.”, Line 168 in the revised manuscript.

  1. “Yellow sticky traps with lure cores (The West Company, Phoenixville, PA, USA)”. Please describe the traps better as I can’t find this company. Flat panels? What dimensions? How arrayed in the field? What is a lure core? How many traps per treatment were deployed in each of the three (or four) repetitions in each field?

Response:

Added the above mentioned points as request, the design of the attraction of different formulations of volatiles on B. huonchili was added in Figure 2.

Change “Yellow sticky traps with lure cores (The West Company, Phoenixville, PA, USA)” to “Yellow sticky traps with rubber septa (Pherobio Technology Co.,Ltd, CHN).” Line 175 in the revised manuscript.

The distance between the two lure cores is 20 m, and each treatment is repeated 3 times.

Figure2. The design of the attraction of different formulations of volatiles on B. huonchili. The red circle represents the traps of formula 6, the purple circle represents formula 10, the green circle represents formula 20, and the black circle represents the control. The distance between the two lure cores is 20 m, and each treatment is repeated 3 times.

  1. Duncan’s multiple comparison test is rarely used any more. Is there a reason that was chosen? Tukey’s HSD might be better.

Response:

Corrected as request, Tukey test is used for multiple comparison in our study. Line 195 in the revised manuscript.

The components of the collected volatiles were identified by aligning their mass spectra with the benchmark compounds from the NIST2011 library and the quantification was evaluated using the peak area normalization method [18]. Line 118 in the revised introduction.

  1. Add “tentatively” identify. What database was used for the identifications?

Response: Corrected as request, SPME and GC-MS were used to identify 25 volatile compounds from Astragalus pods during the pod filling period tentatively, including green leaf volatiles (GLVs), terpenoid and aromatic compounds (Table 3).

  1. What is granulation stage? Earlier you called it the seed-swelling stage. you use the term “drum granule stage”. Please be consistent and perhaps cite a paper that describes the vegetative/reproductive stages of this plant.

Response: The sage of the pods used in our study was pod filling period. We re-read our manuscript carefully and have modified the sage of the pods to “pod filling period” [22]. Line 201 in the revised manuscript.

References:

  1. Mduruma, Z.O.; Nchimbi-Msolla, S.; Reuben, S. O. W. M.; Misangu, R. N. Evaluation of maturity characteristics and of yield components. Tanzania Journal of Agricultural Sciences. 1998. 1: 131-140.

  1. Fig. 1 – what was the identity of the 25 volatiles found in the seed-swelling flower headspace? Please put in a table.

Response: Added the above mentioned points as request, GC-MS analyses identified 25 components in A. membranaceus pods with the relative proportions in the collections summarized in Table 3. Line 208 in the revised manuscript.

  1. Fig. 2- how would you explain that high dosages of hexyl acetate (500 ug/uL) were attractive but low doses (1 and 10 ug/ul) were repellent)? That doesn’t make intuitive sense.

Response: The concentrations of volatiles that are important in determining whether a positive behavioral response is elicited. In Li's study, compared with the control, Chouioia cunea Yang was not attracted to nonadecane at high and low doses, whereas C. cunea was weakly attracted to nonadecane an intermediate dose [21]. In Goelen's study, Aphidius colemani had a significant preference for benzaldehyde at 50 ng and 10 ng doses and a negative response in A. colemani at the higher dose of 1 μg [28]. Therefore, the optimal concentrations of eight compounds, which are commonly found in plants and are easily obtainable, making them feasible as plant-derived at-tractants, were determined to attract B. huonchili wasps. . Line 282 in the revised manuscript.

References:

  1. Li, M.; Yang, Y. X.; Yao, Y. H.; Xiang, W. F.; Han, J. Y.; Wang, Y. H.; Bai, P. H.; Wang, J.; Zhu, G. P.; Man, L.; Zhang, F.; Pan, L. N. Isolation and identification of attractants from the pupae of three lepidopteran species for the parasitoid Chouioia cunea Yang. Pest Management Science. 2020. 76: 1920-1928.
  2. Goelen, T.; József. V.; Sobhy, I. S.;Wackers, F.;John, C. C .; Michael, A. B.; Hans, R.; Hans, J.;Bart, L.Identification and application of bacterial volatiles to attract a generalist aphid parasitoid: from laboratory to greenhouse assays. Pest management science. 2021. 77: 930-938

  1. How did you decide on the specific concentrations of components within the multi-component mixtures?

Response: To investigate the attraction of multi-component formulations to B. huonchili wasps, we selected five kinds of volatiles with high attraction, including 10 µg/µL ocimene, 500 µg/µL hexyl acetate, 100 µg/µL hexanal, 1 µg/µL decanal, and 10 µg/µL β-caryophyllene. The contents and proportion of multi-component formulations were according to the GC-MS results. See the section “2.4.2. Multi-Component Behavioral Assay ” of the text.

  1. l. 213 – at what growth stage were the Astragalus fields used in the field trials? Were they at the seed-swelling stage? If vegetative rather than reproductive, they may have been significantly less attractive than the baited traps. How might the traps compete in fields that have high numbers of attractive seed-swelling stage plants?

Response:

The sage of the pods used in our study was pod filling period.

We think that the reason for the traps compete with pod filling period plants for wasps is that the multi-component formulations of the traps were selected strictly by three steps. Firstly, our results found that specific concentrations of ocimene, n-hexyl acetate, hexanal, decanal, and β-caryophyllene identified from A. membranaceus pods, could attract B. huonchili wasps. Secondly, our findings revealed that the multi-component formulations composed by the five volatiles, such as formulation 6, 10 and 20, exhibited strong attraction to B. huonchili in Y-tube olfactometer assays. Lastly, field insect-attracting evaluations confirmed that Formulation 6 was significantly more effective than the other three formulations. Therefore, the traps of the multi-component formulations have more attraction than the pods. Because the multi-component formulations were composed of the attractive compounds, however, the volatiles of pods contained attractive compounds and repellent compounds, such as 1-octen-3-ol (Figure 2F). Line 315 in the revised manuscript.

  1. Table 3 – please put in the number of replicates and the meaning of the means separations letters.

Response: Mean values in a row without a common letter are significantly different, as determined by a one-way ANOVA followed by Tukey’s HSD test (P <0.05). Each treatment was repeated three times. Line 258 in the revised manuscript.

  1. Discussion – “This suggests that the plant volatiles that attract these five species of Bruchophagus wasps are the same or similar, indicating relatively conserved olfactory genes of these seed wasps and potentially similar functions.” Without knowing the number of each species that you caught, and their true numbers in the field, it is difficult to know if you attracted the five species in similar proportions to their abundance.

Response: We agree with your point that our manuscript should focused on the species of Bruchophagus huonchili, because B. huonchili. is pest that poses a serious threat to the yield and quality of A. membranaceus seeds. Therefore, the result of the field trapping experiment for the three tested formulas is only presented the number of B. huonchili in Table 4.

  1. “…B. huonchili wasps, in combination with sex pheromones”. Has the sex pheromone of this seed wasp been identified? Is there known attraction to a pheromone? If not, this is not really pertinent to the discussion. You discuss attraction of parasitoids by sex pheromone to herbivores but the species you are working on is an herbivore, not a parasitoid. I think the discussion should be more focused on your plant-insect or related plants and related pest insect systems.

Response: We agree with your point that the discussion about sex pheromones of B. huonchil was removed.

Reviewer 2 Report

The Manuscript titled " Deciphering the chemical fingerprint of Astragalus: volatile
components attractive to seed wasps and implications for pest management
", evaluated the influence of volatiles from Astragalus pods in the behavioral response of the wasp Bruchophagus huonchili in laboratory and field conditions. The topic is important to understand the chemical communication of wasps and their host plants and in an applied viewpoint of chemical ecology. However, the study has some gaps that need to be solved before being considered for publication. One of my main concern is about the poor description of the chemical analysis and how solutions concentracions were chosen and criteria used to prepare the blends.  Below specific comments.

Lines 46-52: Please insert references that support these sentences.

Lines 62-65: These lines are in a different color.

Line 101: PDMS.

Lines 109-114: Why were those compounds selected over the other 18 compounds? And how the compounds were identified?

Line 116: Why did authors use a Y-tube experiment? The wasps are flying insects, it would be desirable to perform a wind tunnel bioassay.

How did authors explain the high percentage of non-responding insects?

Lines 116-140: Please, mention in this part about the concentrations used in the single. compounds’ bioassays. And also, justify the election of these concentrations.

Line 145: Why did authors use the proportion of the GC for the preparation of the blend? Is there any biological reason? Please, consider the limitations of the SPME technique in the semi-quantitative analysis (https://onlinelibrary.wiley.com/doi/full/10.1002/ffj.1991) and (https://link.springer.com/article/10.1007/s10886-009-9733-2)

Did the authors do a calibration curve?

Figure 1: The chromatogram is not suitable, it is better to present a table with the details of the chemical identification performed. I would like to ask the authors to provide the RI and chemical identification of the compounds.

Figure 2: Please delete the sign minus “-“ from the negatives. Please avoid using the percentages and use the number of insects choosing each treatment. Additionally, add the non-responders in the Figure.

Line 213, why did authors test only formulations (blends) and did not test individual compounds in field bioassays? What is the justification for making blends of compounds that are individually attractive?.

For example, ocimene and hexanal attract 28 and 26 percent of the insects, the same percentages of the formulations 20, 10 and 6.

It would be cheaper and easier to use one compound than a blend.

Line 294: change synthetized formulas for blends.

Discussion should be improved.

Other comments and revisions are described in the PDF file attached.

Moderate editing of English language is required

Author Response

Comments to the Author

  1. Please insert references that support these sentences.

Response: There are still no registered pesticides on the Chinese Pesticide Information Network(http://www.chinapesticide.org.cn/) that can be used for pest control of Astragalus membranaceus. Moreover, there is a potential risk of pesticide residues in the application of pesticides, which affects the medicinal value of A. membranaceus. Therefore, finding an effective, feasible, and environmentally friendly measure to control Astragalus seed wasps is of great significance for the sustainable development of the Astragalus industry.

  1. These lines are in a different color.

Response: Change the color of these lines to black.

  1. Line 101: PDMS.

Response: Change “PDM” to “PDMS”.

  1. Why were those compounds selected over the other 18 compounds? And how the compounds were identified?

Response:

The components of the collected volatiles were identified by aligning their mass spectra with the benchmark compounds from the NIST2011 library and the quantification was evaluated using the peak area normalization method [18]. Line 118 in the revised introduction.

The reason for selection the 8 compounds from the 25 volatile compounds found in Astragalus membranaceus pods is according to the high content and the function in the interaction of insect and host plant.

  1. Why did authors use a Y-tube experiment? The wasps are flying insects, it would be desirable to perform a wind tunnel bioassay.

Response: The individual size of Bruchophagus huonchili is only 2.3-2.5mm, so it is suitable for Y-tubes. Moreover, Y-tube experiment is employed for study the behavior of wasps, such as Chouioia cunea Yang and Campoletis chlorideae [1-2].

References:

  1. Li, M.; Yang, Y. X.; Yao, Y. H.; Xiang, W. F.; Han, J. Y.; Wang, Y. H.; Bai, P. H.; Wang, J.; Zhu, G. P.; Man, L.; Zhang, F.; Pan, L. N. Isolation and identification of attractants from the pupae of three lepidopteran species for the parasitoid Chouioia cunea Yang. Pest Management Science. 2020. 76: 1920-1928.
  2. Sun, Y. L.; Dong, J. F.; Ning, C.; Ding, P. P.; Huang, L. Q.; Sun, J. G.; Wang, C. Z. An odorant receptor mediates the attractiveness of cis-jasmone to Campoletis chlorideae, the endoparasitoid of Helicoverpa armigera. Insect Molecular Biology. 28: 23-34.

  1. How did authors explain the high percentage of non-responding insects?

Response:  The data shown in the Figure 3 is the number of selected wasps, which means parasitoids that chose the treatment arm and the control arm. Meanwhile, the rate of unselected wasps is less than 20%.

  1. Lines 116-140: Please, mention in this part about the concentrations used in the single. compounds’ bioassays.

Response: The concentrations used in the single compounds’ bioassays has been described in the section of 2.4 “Test solutions were prepared at concentrations of 1, 10, 100, and 500 µg/µL”.  Line 147 in the revised manuscript. According to the references of other studies, doses of 0.01 mg ~ 5 mg were selected to evaluate the response to B. huonchili  [1-2].

References:

  1. Li, M.; Yang, Y. X.; Yao, Y. H.; Xiang, W. F.; Han, J. Y.; Wang, Y. H.; Bai, P. H.; Wang, J.; Zhu, G. P.; Man, L.; Zhang, F.; Pan, L. N. Isolation and identification of attractants from the pupae of three lepidopteran species for the parasitoid Chouioia cunea Yang. Pest Management Science. 2020. 76: 1920-1928.
  2. Roberts, J. M.; Kundun, J.; Rowley, C.; Hall, D.;. Paul, D.; Tom, P. W. Electrophysiological and Behavioral Responses of Adult Vine Weevil, Otiorhynchus sulcatus (Coleoptera: Curculionidae), to Host Plant Odors. Journal of Chemical Ecology. 2019. 45 : 858-868.

8.Why did authors use the proportion of the GC for the preparation of the blend? Is there any biological reason? Please, consider the limitations of the SPME technique in the semi-quantitative analysis (https://onlinelibrary.wiley.com/doi/full/10.1002/ffj.1991) and (https://link.springer.com/article/10.1007/s10886-009-9733-2)Did the authors do a calibration curve?

Response: Firstly, the proportion of the GC is the actual ratio of the volatiles of the pods. Therefore, we thought the ratio of the volatiles according to the GC should be more effective to B. huonchili wasps. Similar to previous study, Wang found that the ratio of 2-hexenal and benzaldehyde attracted more Callosobruchus chinensis females than other ratios, suggesting that the ratio of the volatile blends based on the GC-MS peak areas are important for herbivore attraction [1]. Secondly, although the limitations of the SPME technique, SPME was used to identify the volatiles of plant in many studies, such as pods of Vigna radiate, tea plants, and so on [1-2]. Lastly, according to the doses of different volatiles, we could make sure the effective doses for B. huonchili wasps. Therefore, we did not made the calibration curve.

References:

  1. Wang, H. M.; Bai, P. H.; Zhang, J.; Zhang, X. M.; Hui, Q.; Zheng, H. X.; Zhang, X. H. Attraction of bruchid beetles Callosobruchus chinensis (L.) (Coleoptera: Bruchidae) to host plant volatiles. Journal of Integrative Agriculture. 2020. 19: 2-11.
  2. Jing, T. T.; Du, W. K.; Gao, T.; Wu,Y.; Zhang, N.; Zhao, M. Y.; Jin, J. Y.;Wang, J. M.;Wilfried, S.; Wan, X. C.; Song, C. K. Herbivore-induced DMNT catalyzed by CYP82D47 plays an important role in the induction of JA-dependent herbivore resistance of neighboring tea plants. Plant, cell & environment. 2020. 44: 1178-1191.

  1. Figure 1: The chromatogram is not suitable, it is better to present a table with the details of the chemical identification performed. I would like to ask the authors to provide the RI and chemical identification of the compounds.

Response: Added the above mentioned points as request, GC-MS analyses identified 25 components in A. membranaceus pods with the relative proportions in the collections summarized in Table 3. Line 208 in the revised manuscript.

  1. Figure 2: Please delete the sign minus “-“ from the negatives. Please avoid using the percentages and use the number of insects choosing each treatment. Additionally, add the non-responders in the Figure.

Response: As required, we have deleted the sign minus “-“ from the negatives.

The number in the bar indicates the total number of wasps choosing the arm. The black bar represents paraffin oil treatment (control) and the white bar represents the tested single compounds. Because the total number of wasps in each treatment was different, so we think that the percentage of wasps for each compound was better for expressing the attraction than the number. Additionally, the non-responders the rate of non-responders is less than 20%, which has been added in Line 223 in the revised manuscript.

  1. why did authors test only formulations (blends) and did not test individual compounds in field bioassays? What is the justification for making blends of compounds that are individually attractive?

Response: Firstly, our results found that specific concentrations of ocimene, n-hexyl acetate, hexanal, decanal, and β-caryophyllene showed attraction to B. huonchili wasps with 67.65% (c2=4.235, P=0.040), 67.74% (c2=3.903, P=0.048), 65.12% (c2=3.903, P=0.047), 67.57% (c2=4.568, P=0.033), and 66.67% (c2=4.000, P=0.046) in laboratory, respectively (Fig. 3). Secondly, our findings revealed that the multi-component formulations composed by the five volatiles, such as formulation 6, 10 and 20, exhibited strong attraction to B. huonchili in Y-tube olfactometer assays, with the highest being 73.07% for formula 6 (c2=7.410, P=0.006), followed by formulas 10 (c2=4.333, P=0.037) and 20 (c2=3.930, P=0.047), with attraction of 66.66% and 65.12%, respectively (Fig. 4). Overall, formula 6 is the most effective among the test individual compounds and synthetized formulas. Moreover, perception of blends of plant volatiles, rather than individual components, plays a pivotal role in host recognition. Therefore, formulation 6, 10 and 20 were confirmed in field insect-attracting evaluations, and. Therefore, the traps of the multi-component formulations have more attraction than the pods. Because the multi-component formulations were composed of the attractive compounds, with formula 10 and formula 20 as positive controls.

  1. For example, ocimene and hexanal attract 28 and 26 percent of the insects, the same percentages of the formulations 20, 10 and 6.

Response: In our laboratory experiment, it was found that the attraction rate of formula 6 was 73.07%, significantly higher than that of single component ocimene and hexanal. Moreover, perception of blends of plant volatiles, rather than individual components, plays a pivotal role in host recognition. In addition, field insect-attracting evaluations confirmed that Formulation 6 was significantly effective to B. huonchili wasps (p<0.05). Although we did not test individual compounds in field bioassays, but we thought the effect of formulation 6 was better than individual compounds.

14.It would be cheaper and easier to use one compound than a blend.Line 294: change synthetized formulas for blends.

Response: Although using a single compound is cheaper and easier, the lure rate of a single component is significantly lower than that of Formula 6.

  1. Line 294: change synthetized formulas for blends.

Response: This discussion has been deleted.

Reviewer 3 Report

In this manuscript, Zhang et al. identified throughout GC-MS volatile components released from Astragalus pods and performed attraction olfactometry bioassays on Bruchophagus huonchili wasps, either using mono components or with synthetic formulations. The best three formulations were assayed on the Astragalus plant fields showing efficiency on other Bruchophagus species.

Despite the research methodology is appropriate, in my opinion, many items must be addressed before publication.

Moreover, this reviewer considers that, considering the promising results obtained, the discussion section is too poor and scarcely related to the biology of the studied insect and plant.

Please, attend to the following comments:                                     

Collection de volatiles and GC analysis:

Explain the process of the selection of the plant and the pod.

Please provide details on how the volatiles associated with cutting plant material were avoided.

Up to the authors, I would recommend including an image of the pod.

Provide evidence* of the presence of the most relevant components, hexyl acetate, hexanal, decanal, and caryophyllene. *Can be with a table that includes retention times, relative compositions, and % of coincidence with MS library, or by indicating peak and area on the shown chromatogram.

Line 174. Is it “hexenal” or hexanal?

Behavioral assays:

Since female adult wasps lay their eggs beneath the seed coat inside the pods, I would suggest detailing the process by which the insects were chosen for the experiment. Was there any control between gravid and virgin females? Average age of adults?

Prior to trials with single component trials or formulations, was the Astragalus pod tested?

To show that there are no biases in the design of the experiment (light, olfactometer, heat), shouldn't there be a control experiment -made with both arms with paraffin- on every result from figure 2? or a separate experiment was done?

Figure 2 D, Decanal 1 ug/uL. Please double check the total number of insects at each arm.

Table 3. The meaning of a b c d is missing.

Field trapping experiments:

If it is possible, can the authors add images of the traps disposed of on the field before and after field experiment? Were the captured insects classified by species, sex, or gravid?

Author Response

1.Collection de volatiles and GC analysis:

Explain the process of the selection of the plant and the pod. Please provide details on how the volatiles associated with cutting plant material were avoided.

Response: During the pod filling period, the whole plants of three year old A. mongolicus has been taken from the field to our lab. The health pods were collected from the plants and a total weight of 2 g pods were prepared for volatile extraction immediately. Line 109 in the revised introduction

  1. Up to the authors, I would recommend including an image of the pod.

Response: Added pod photos in section 2.1 of the text. For example,

Figure 1. The pods of A. membranaceus during pod filling period in the field.

  1. Provide evidence* of the presence of the most relevant components, hexyl acetate, hexanal, decanal, and caryophyllene. *Can be with a table that includes retention times, relative compositions, and % of coincidence with MS library, or by indicating peak and area on the shown chromatogram.

Response: Added the above mentioned points as request, GC-MS analyses identified 25 components in A. membranaceus pods with the relative proportions in the collections summarized in Table 3. Line 208 in the revised manuscript.

  1. Is it “hexenal” or hexanal?

Response: Change “hexenal” to “hexanal”.

  1. Since female adult wasps lay their eggs beneath the seed coat inside the pods, I would suggest detailing the process by which the insects were chosen for the experiment. Was there any control between gravid and virgin females? Average age of adults?

Response: The pupae of the species for behavioral assays were B. huonchili. Newly emerged wasps (0-24 h) were isolated and identify the species of B. huonchili according to the key characteristics[19]. Line 101 in the revised introduction.

References:

  1. Fan, Y.; Yang, CQ.; Lv, X. M. Study on Bruchophagus huonchili. Chinese Entomology Towards the 21st Century-Proceedings of the 2000 Academic Annual Academic Conference of the Chinese Entomology Society, Yichang. 2000. 1122-1126.

  1. Prior to trials with single component trials or formulations, was the Astragalus pod tested?

Response: It is important to test the attraction of the pods of A. membranaceus to B. huonchili wasps. However, we did not conduct the experiment. Because it is common that the host plants could attract insects by their volatiles, and the phenomenon can be observed in the fields of A. membranaceus. Therefore, we did not test the the attraction of the pods of A. membranaceus to B. huonchili wasps.

  1. To show that there are no biases in the design of the experiment (light, olfactometer, heat), shouldn't there be a control experiment -made with both arms with paraffin- on every result from figure 2? or a separate experiment was done?

Response: Added the above mentioned points as request, “Before assessing the attraction of individual and combined compounds to B. huonchili, a control experiment were conducted. In the control experiment, 10 μL of paraffin oil was applied to a filter paper strip and placed in each arm. The results showed that B. huonchili wasps distributed equivalent numbers in each arm in the absence of plant volatiles, which demonstrated that the Y-tube olfactometer was suitable to measure the response rates of B. huonchili, to different volatiles and component formulations”. Line 140 in the revised manuscript.

  1. Figure 2 D, Decanal 1 ug/uL. Please double check the total number of insects at each arm. Table 3. The meaning of a b c d is missing.

Response: We have checked the total number of insects at each arm and revised the number. Added the above mentioned points as request, mean values in a row without a common letter are significantly different, as determined by a one-way ANOVA followed by Tukey’s HSD test (P <0.05).

  1. If it is possible, can the authors add images of the traps disposed of on the field before and after field experiment? Were the captured insects classified by species, sex, or gravid?

Response: Firstly, we have added images of the traps disposed of on the field before and after field experiment. (Figure5).

Figure 5 Field trapping effects of synthetized formulas on B. huonchili wasps.

(A) The images of the traps disposed of on the field before the field experiment,

(B) The images of the traps disposed of on the field after the field experiment.

Secondly, according to the suggestion of another reviewer, we focus on the species of Bruchophagus huonchili, which poses a serious threat to the yield and quality of A. membranaceus seeds. Therefore, the result of the field trapping experiment for the three tested formulas is only presented the number of B. huonchili in Table 4.

Round 2

Reviewer 2 Report

In my PDF revision I requested to state wich ocimene isomer you found.  I can not see  this correction.  I understand that you used  ocimene ( a mixture of isomers) in the bioassays which it is valid but you have to mention, but,  in the list of compouds you should state if you found the  beta-cis-ocimene or beta-trans-ocimene. It is important.

Author Response

  1. In my PDF revision I requested to state which ocimene isomer you found. I can not see this correction. I understand that you used ocimene ( a mixture of isomers) in the bioassays which it is valid but you have to mention, but, in the list of compouds you should state if you found the beta-cis-ocimene or beta-trans-ocimene. It is important.

Response: We are sorry for neglecting this question, and we have changed “ocimene” to “cis-β- ocimene” in the revised manuscript.